# A Combination of an Angiotensin II Receptor and a Neprilysin Inhibitor Attenuates Liver Fibrosis by Preventing Hepatic Stellate Cell Activation

**DOI:** 10.3390/biomedicines11051295

**Published:** 2023-04-27

**Authors:** Junya Suzuki, Kosuke Kaji, Norihisa Nishimura, Takahiro Kubo, Fumimasa Tomooka, Akihiko Shibamoto, Satoshi Iwai, Yuki Tsuji, Yukihisa Fujinaga, Koh Kitagawa, Tadashi Namisaki, Takemi Akahane, Hitoshi Yoshiji

**Affiliations:** Department of Gastroenterology, Nara Medical University, Kashihara 634-8521, Nara, Japan

**Keywords:** liver fibrosis, hepatic stellate cell, natriuretic peptide, angiotensin II

## Abstract

The renin–angiotensin–aldosterone system has gained attention due to its role as a mediator of liver fibrosis and hepatic stellate cell (HSC) activation. Meanwhile, the natriuretic peptide (NP) system, including atrial NP (ANP) and C-type NP (CNP), is a counter-regulatory hormone regulated by neprilysin. Although the combination of an angiotensin receptor and a neprilysin inhibitor (sacubitril/valsartan: SAC/VAL) has shown clinical efficacy in patients with heart failure, its potential effects on hepatic fibrosis have not been clarified. This study assessed the effects of SAC/VAL in carbon tetrachloride (CCl_4_)-induced murine liver fibrosis as well as the in vitro phenotypes of HSCs. Treatment with SAC and VAL markedly attenuated CCl_4_-induced liver fibrosis while reducing α-SMA^+^-HSC expansion and decreasing hepatic hydroxyproline and mRNA levels of pro-fibrogenic markers. Treatment with SAC increased plasma ANP and CNP levels in CCl_4_-treated mice, and ANP effectively suppressed cell proliferation and TGF-β-stimulated *MMP2* and *TIMP2* expression in LX-2 cells by activating guanylate cyclase-A/cGMP/protein kinase G signaling. Meanwhile, CNP did not affect the pro-fibrogenic activity of LX-2 cells. Moreover, VAL directly inhibited angiotensin II (AT-II)-stimulated cell proliferation and the expression of *TIMP1* and *CTGF* through the blockade of the AT-II type 1 receptor/protein kinase C pathway. Collectively, SAC/VAL may be a novel therapeutic treatment for liver fibrosis.

## 1. Introduction

Liver fibrosis is a common pathological feature in chronic liver diseases secondary to a spectrum of etiologies, including hepatitis B and C, excessive alcohol consumption, and non-alcoholic fatty liver disease [1,2,3]. Advanced liver fibrosis leads to portal hypertension, ascites, and variceal bleeding, which ultimately lead to hepatocarcinogenesis [4,5,6]. Hepatic stellate cells (HSCs) largely contribute to the progression of liver fibrosis. In the healthy liver, HSCs remain at a quiescent state and are responsible for vitamin A storage in cytoplasmic lipid droplets [7]. Upon persistent hepatic injury, HSCs are activated and trans-differentiate into myofibroblasts that generate extracellular matrix (ECM), resulting in an imbalance between matrix metalloproteinases (MMPs) and tissue inhibitors of metalloproteinases (TIMPs) [8]. Therefore, one of the most critical targets for anti-fibrotic therapy is the prevention of HSC proliferation and activation.

Neurohumoral vasoconstrictor systems play a key role in HSC activation. Particularly, the renin–angiotensin–aldosterone system (RAAS) modulates the proliferative capacity and fibrogenic activity of HSCs through the pro-fibrotic effector angiotensin II (AT-II) [2,9,10,11]. AT-II induces HSC proliferation, migration, collagen production, and the secretion of transforming growth factor (TGF)-β1 and pro-inflammatory mediators [12,13]. Moreover, the pharmacological blockade of AT-II activity by the angiotensin-converting enzyme inhibitor (ACE-I) or AT-II receptor type 1 (AT1R) blocker (ARB) suppresses the pro-fibrotic properties of HSCs and attenuates liver fibrosis in rodents [14,15,16]. However, therapeutic strategies for preventing fibrosis using ACE-I and ARB alone in clinical settings have been unsuccessful.

The natriuretic peptide (NP) system is also recognized as another critical counter-regulatory pathway that maintains hemodynamic homeostasis [17]. The NP system includes atrial NP (ANP), B-type NP, and C-type NP (CNP) that act as ligands for receptors that are membrane-bound to guanylyl cyclases (GC), thereby increasing intracellular levels of cyclic guanosine monophosphate (cGMP) and its effector, protein kinase G (PKG) [18,19]. Several studies have suggested that NPs may have an inhibitory effect on HSC activation. A previous rodent study indicated that continuous infusion of ANP suppressed liver fibrosis and HSC activation [20]. Furthermore, CNP also has the potential to inhibit the growth of myofibroblastic HSCs [21].

A vital regulator of the NP system is neutral endopeptidase neprilysin (NEP), which catalyzes NPs as well as degrading bradykinin, endothelin-1, and AT-II [22]. Accordingly, NEP inhibition not only augments the bioactivity of the NP system but also activates RAAS. Thus, sacubitril (SAC), which is a NEP inhibitor (NEP-i), is combined with an ARB (valsartan: VAL) to serve as an AT-II receptor and NEP inhibitor (ANRI) for the treatment of patients with heart failure with reduced ejection fraction [23]. Based on the functional properties of AT-II and NPs, ANRI might have therapeutic potential against the fibrogenic activity of HSCs and prevent hepatic fibrosis. 

This study examined the preventive effects of combined NEP-i with ARB in vivo in carbon tetrachloride (CCl_4_)-induced murine liver fibrosis and the in vitro activation of HSCs.

## 2. Materials and Methods

### 2.1. In Vivo Experimental Design

Eight-week-old male C57BL/6NCrSlc mice (Japan SLC, Inc., Hamamatsu, Japan) were caged with free access to food and water and kept at 23 ± 3 °C with 50 ± 20% humidity and a 12 h light/dark cycle. Overall, 50 mice were randomly divided into 5 experimental groups (*n* = 10) and underwent treatment for 8 weeks as follows (Figure 1A): (i) intraperitoneal injection of corn oil and oral gavage of saline as a vehicle (C/O group); (ii) intraperitoneal injection of CCl_4_ (FUJIFILM Wako Pure Chemical Corporation, Osaka, Japan) twice a week (1 mL/kg) and vehicle (CCl_4_ + Veh group); (iii) CCl_4_ and oral gavage of sacubitril (30 mg/kg) (Tokyo Chemical Industry, Tokyo, Japan) (CCl_4_ + SAC group); (iv) CCl_4_ and oral gavage of valsartan (30 mg/kg) (Tokyo Chemical Industry) (CCl_4_ + VAL group); and (v) CCl_4_ and oral gavage of sacubitril and valsartan (30 mg/kg each) (CCl_4_ + SAC/VAL group) [24,25,26]. After 8 weeks, all of the mice were intravenously injected with pentobarbital sodium (150 mg/kg), subjected to anesthesia, and then their blood and livers were collected immediately after euthanasia. Serum levels of biochemical markers were assessed at SRL, Inc. (Tokyo, Japan). 

### 2.2. Ethics of the Animal Study

All of the studies were approved, and the methods were performed in accordance with the National Institutes of Health Guide for the Care and Use of Laboratory Animals (NIH publications number 80–23) revised in 2011, and all of the experimental procedures were approved by the Animal Ethics Committee of Nara Medical University (authorization number: 12910).

### 2.3. Mouse Plasma AT-II, ANP, and CNP Measurement

Plasma levels of AT-II, ANP, and CNP in all of the mice were measured using an Angiotensin II ELISA kit (Enzo Life Science, Farmingdale, NY, USA), a Mouse Atrial NP/ANP enzyme-linked immunosorbent assay (ELISA) Kit (Novus Biologicals, Centennial, CO, USA), and a Mouse CNP ELISA Kit (Innovative Research, Inc., Novi, MI, USA), respectively, according to the manufacturer’s protocols.

### 2.4. Histological and Immunohistochemical Analyses

Liver and gastrocnemius muscle tissues were fixed in 10% formalin for 24 h at room temperature. The paraffin-embedded specimens were sectioned at 5 μm and subjected to hematoxylin and eosin (H&E) and Sirius Red staining (performed at Narabyouri Research Co. Ltd., Nara, Japan). Rabbit-polyclonal alpha-smooth muscle actin (α-SMA) (ab5694, 1:50, Abcam) was used as the primary antibody for immunostaining. Goat anti-rabbit IgG (H + L) and horseradish peroxidase (HRP) (1:1000, Thermo Fisher Scientific, Waltham, MA, USA) were used as the secondary antibody and visualized using an HRP-conjugated ABC system (Vector Laboratories, Burlingame, CA, USA). 3,3’-Diaminobenzidine was used as a chromogen for visualization. Semi-quantitative analysis was performed using ImageJ 64-bit Java 1.8.0. (National Institutes of Health, Bethesda, MD, USA).

### 2.5. Cell Culture

A human HSC line, LX-2 (Merck, Darmstadt, Germany), was cultured in 2% fetal bovine serum (Thermo Fisher Scientific) containing DMEM (Nacalai Tesque, Kyoto, Japan) supplemented with 1% penicillin/streptomycin at 37 °C and 5% CO_2_. The cells were incubated with recombinant human ANP (Abcam, Cambridge, UK), recombinant human CNP (Novus Biologicals), recombinant TGF-β1 (Sigma-Aldrich, St. Louis, MO, USA), and/or a PKG inhibitor (KT5823; Abcam). For another set of assays, the cells were exposed to human AT-II (Sigma-Aldrich) and VAL. 

### 2.6. Cell Proliferation Assay

The LX-2 cells were cultured on uncoated plastic dishes at a density of 15,000 cells/mL and exposed to ANP (0–500 pg/mL), CNP (0–500 pg/mL), or AT-II (10^−8^–10^−5^ M) or different durations (0–48 h). To assess the effects of inhibition of PKG or AT1R, KT5823 (10 μM) or VAL (10^−7^–10^−5^ M) was added to the cells after 30 min of treatment with/without ANP (0, 100, and 500 pg/mL) or with/without AT-II (10^−6^ M) [27]. Cell proliferation was assessed by using the Premix WST-1 Cell Proliferation Assay system (Takara Bio, Kusatsu, Japan). Three independent experiments were performed, and the mean results were calculated from five replicates of each experiment. The proliferative rate was indicated as the ratio to the group without treatment at the start of each experiment. 

### 2.7. Intracellular cGMP Assay

The intracellular cGMP level was measured using a cGMP complete ELISA kit (Enzo Life Science) according to the manufacturer’s protocol. For the assays, 1 × 10^6^ of LX-2 cells were exposed to ANP (0–1000 pg/mL) or CNP (0–1000 pg/mL) for 15 min.

### 2.8. PKG Activity Assay

Intracellular PKG activity was determined using a Human PKG/PRKG1 ELISA Kit (LSBio, Seattle, WA, USA). Briefly, 1 × 10^6^ of LX-2 cells were treated with ANP (0–1000 pg/mL) or CNP (0–1000 pg/mL) for 30 min, and the pelleted cells underwent centrifugation to remove the supernatant. After cell lysis by ultrasonication, the supernatant was applied for the ELISA assay. The optical density was measured using a microplate reader to 450 nm.

### 2.9. Protein Kinase C (PKC) Activity Assay

Intracellular PKC activity was measured using the PKC Kinase Activity Assay Kit (Abcam) according to the manufacturer’s instructions. For the assay, 1 × 10^6^ of LX-2 cells were treated with AT-II (10^−8^–10^−5^ M) for 30 min. The protein concentration of the samples was determined with bicinchoninic acid to normalize PKC activities to the protein concentrations.

### 2.10. RNA Isolation and Real-Time Quantitative PCR 

Total RNA isolation was performed using RNeasy Mini kits (Qiagen, Hilden, Germany) for mouse liver tissues and cultured LX-2 cells. Reverse transcription was performed using high-capacity RNA-to-cDNA kits (Thermo Fisher Scientific). Real-time qPCR was performed using an SYBR™-Green PCR Master Mix (Thermo Fisher Scientific) and an Applied Biosystems StepOnePlus™ Real-Time PCR^®^ system (Thermo Fisher Scientific). The primer pairs are listed in Appendix A. The relative expression of each gene was normalized to glyceraldehyde-3-phosphate dehydrogenase (GAPDH) expression and estimated using the 2^−ΔΔCq^ method [28]. The expression levels were presented as the fold-change relative to the control.

### 2.11. Western Blotting Assay

Whole-cell lysates were extracted from cultured LX-2 cells using freshly prepared RIPA lysis buffer (Sigma-Aldrich), containing a proteinase and phosphatase inhibitors cocktail (Thermo Fisher Scientific). A Bradford colorimetric assay (Bio-Rad, Hercules, CA, USA) was used to measure the concentration of the protein. Equal amounts of protein (50 µg) were separated on SDS-PAGE and transferred onto PVDF membranes. After blocking with 5% skimmed milk, the blots were incubated overnight at 4 °C with the following primary antibodies: rabbit-polyclonal anti-natriuretic peptide receptor (NPR)1 antibody (1:1000, PA5-29049, Thermo Fisher Scientific); rabbit-polyclonal anti-NPR2 antibody (1:1000, PA5-99836, Thermo Fisher Scientific); and rabbit-monoclonal anti-GAPDH (1:1000, 2118, Cell Signaling Technology, Danvers, MA, USA). The membranes were then washed thrice and incubated with goat anti-rabbit IgG H&L (HRP) (1:2000, ab6721, Abcam) for 1 h at room temperature. The blots were developed with a Clarity Western ECL Substrate (Bio-Rad) using an iBright™ CL1500 Imaging System (Thermo Fisher Scientific). 

### 2.12. Statistical Analyses

Data are presented as the mean ± standard deviation. A two-tailed Student’s t-test and one-way ANOVA followed by Bonferroni’s post hoc test were used to analyze statistical differences. *p* < 0.05 was considered statistically significant.

## 3. Results

### 3.1. Combination of NEP-i with ARB Suppresses CCl_4_-Induced Liver Fibrosis

The experimental design is shown in Figure 1A. Significant body weight loss was observed in the CCl_4_ + Veh group which did not change in the CCl_4_ + SAC and CCl_4_ + VAL groups (Figure 1B). The CCl_4_ + Veh group showed marked hepatomegaly, which was attenuated in the CCl_4_ + VAL group but not in the CCl_4_ + SAC group (Figure 1C). We next measured the effects of both agents on hepatic inflammation. Serological analysis indicated the elevation of aspartate transaminase and alanine transaminase in the CCl_4_ + Veh group which did not change in the CCl_4_ + SAC and CCl_4_ + VAL groups (Figure 1D,E). H&E staining also showed that there was no significant difference in the infiltration of inflammatory cells among all groups (Figure 1F). These findings indicate that neither NEP-i nor ARB enhances CCl_4_-induced hepatic inflammation as well as the fact that both agents did not cause hepatotoxicity at the dose used in this study. Additionally, treatment with SAC and VAL did not cause renal damage in the respective treatment groups (Figure 1G,H). 

Next, we investigated the effects of NEP-i and ARB on CCl_4_-induced liver fibrosis. As shown in Figure 2A,B, continuous CCl_4_ administration for 8 weeks resulted in the development of fibrosis as visualized by Sirius Red staining. Liver fibrosis was significantly attenuated in the CCl_4_ + SAC and CCl_4_ + VAL groups as compared with the CCl_4_ + Veh group (Figure 2A,B). Additionally, both treatments with SAC and VAL reduced the number of α-SMA-positive myofibroblasts, which was increased in the CCl_4_ + Veh group (Figure 2A,C). Notably, the anti-fibrotic effect was greater in the CCl_4_ + SAC/VAL group as compared with the groups treated with either SAC or VAL alone (Figure 2A–C). Furthermore, hepatic levels of hydroxyproline decreased in the CCl_4_ + SAC and CCl_4_ + VAL groups; however, the decrease was greater in the CCl_4_ + SAC/VAL group (Figure 2D). The mRNA expressions of pro-fibrotic genes (*ACTA2*, *COL1A1*, and *TGF-β1*) were also markedly decreased in the livers of the CCl_4_ + SAC, VAL, and SAC/VAL groups (Figure 2E).

### 3.2. NEP-i and ARB Increase the Serum Levels of ANP and CNP in the CCl_4_-Treated Mice

NEP preferentially degrades NPs including ANP and CNP by breaking the ring and inactivating the peptide [29]. Based on this enzymatic property of NEP, we next evaluated the change in plasma levels of ANP and CNP in the experimental mice. Plasma levels of ANP and CNP did not change in the CCl_4_ + Veh group as compared with the C/O group (Figure 3A,B). As expected, plasma levels of both ANP and CNP were elevated in the CCl_4_ + SAC group as compared with the CCl_4_ + Veh group, indicating the effective inhibition of NEP (Figure 3A,B). Meanwhile, there was no significant difference in the plasma levels between the CCl_4_ + Veh and CCl_4_ + VAL groups (Figure 3A,B). We also evaluated the change in the hepatic expression of AT1R and plasma AT-II following treatment with both agents. The CCl_4_ + Veh group showed higher expression of hepatic AT1R as compared with the C/O group. Meanwhile, the CCl_4_-induced increase in hepatic AT1R levels was not observed in the CCl_4_ + SAC and CCl_4_ + VAL groups (Figure 3C). Likewise, treatment with SAC/VAL did not affect the CCl_4_-induced elevation of plasma AT-II levels (Figure 3D).

### 3.3. Increased ANP Levels Inhibit Proliferative and Fibrogenic Activity through cGMP/PKG Signaling

We next assessed the regulatory effects of ANP in HSC proliferation and activation. As shown in Figure 4A, NPR1/GC-A, a receptor of ANP, was present in LX-2 cells whereas NPR2/GC-B, a receptor of CNP, was absent in LX-2 cells in contrast to Hela cells as a positive control. Administration of ANP increased intracellular cGMP levels to concentrations above 100 pg/mL, suggesting ANP-mediated activation of GC-A in LX-2 cells (Figure 4B). In accordance with the increase in cAMP, PKG activity in LX-2 cells was augmented by the administration of ANP (Figure 4C). cGMP-dependent PKG activation was suggested to negatively regulate the cell proliferation of fibroblasts [30]. Additionally, the administration of ANP suppressed the proliferation of LX-2 cells (Figure 4D). Notably, concomitant treatment with KT5823, a PKG inhibitor, significantly negated the inhibitory effects of ANP on cell proliferation (Figure 4E). We also found that ANP suppressed the TGF-β1-stimulated upregulation of *ACTA2* and *COL1A1* expression (Figure 4F,G). Moreover, ANP also repressed the TGF-β1-stimulated upregulation of *MMP2* and *TIMP2* but not *TIMP1* expression (Figure 4H–J). Pharmacological inhibition of PKG signaling effectively inhibited the ANP-mediated suppression of the fibrogenic activity of LX-2 cells (Figure 4E–J). These findings suggest that both the anti-proliferative and anti-fibrogenic effects of ANP are mediated through GC-A/cGMP/PKG signaling. Meanwhile, the administration of CNP did not modify intracellular cAMP and PKG activity (Appendix A). Accordingly, no obvious effects were detected in LX-2 cells following treatment with CNP (Appendix A).

### 3.4. ARB Suppresses AT-II-Induced HSC Proliferation and Pro-Fibrogenic Activity

Finally, we confirmed the direct effects of ARB on AT-II-stimulated HSC proliferation and activation. As shown in Figure 5A, AT-II stimulation enhanced PKC activity in LX-2 cells in a dose-dependent manner, as previously described [31,32]. Treatment with VAL significantly inhibited AT-II-induced PKC activity, indicating that AT-II could act through AT1R/PKC signaling (Figure 5B). According to the PKC activity, AT-II stimulated the proliferative capacity of LX-2 cells in a dose-dependent manner (Figure 5C), and AT-II-induced cell proliferation was efficiently suppressed by VAL (Figure 5D). AT-II stimulation also upregulated the expression of *ACTA2* and *COL1A1* as well as that of *TIMP1* and *CTGF* in LX-2 cells (Figure 5E–H). Meanwhile, treatment with VAL inhibited AT-II-induced upregulation of these genes in correspondence to reduced PKC activity (Figure 5E–H). These findings suggest that ARB directly inhibits the proliferation and pro-fibrogenic activity of HSCs by blocking the AT1R/PKC pathway.

## 4. Discussion

Despite recent advances in pathological knowledge, there are no effective treatment options for hepatic fibrosis. This study has revealed the preventive effects of the dual inhibition of NEP and AT-II by SAC/VAL in CCl_4_-induced liver fibrosis in mice. We also demonstrated that SAC increases plasma ANP levels, which is associated with the inhibition of HSC proliferation and ECM production. Additionally, VAL directly suppresses HSC proliferation and activation in human LX-2 cells.

Our data also show that plasma ANP levels increased following the administration of SAC in CCl_4_-treated mice. ANP is secreted from the atria of the heart and promotes natriuresis, vasodilatation, and endothelial permeability as well as suppressing adrenal aldosterone production [17,18,19]. ANP binds to GC-A, a transmembrane receptor containing an intracellular GC domain, which subsequently synthesizes cGMP and activates PKG [17,18,19,33]. It has been reported that the ANP/cGMP/PKG pathway also plays a key role in delaying tissue fibrosis. Li et al. demonstrated that ANP suppresses TGF-β/Smad signaling and trans-differentiation into myofibroblasts in murine cardiac fibroblasts [34]. Nishikimi et al. suggested that the ANP/GC-A axis exerts inhibitory effects against murine renal fibrosis [35]. Moreover, Ishigaki et al. showed that continuous intravenous administration of ANP ameliorated dimethyl-nitrosamine-induced hepatic fibrosis in rats [20]. Consistent with these reports, our results suggest that increased levels of plasma ANP following treatment with SAC are responsible for the attenuation of hepatic fibrosis in CCl_4_-treated mice. 

Therefore, we targeted the impact of ANP on HSCs as a mechanism underlying the anti-fibrotic effects of SAC. Our results revealed the expression of NPR1/GC-A in human-activated HSCs and LX-2 cells, which is consistent with existing evidence that HSCs isolated from CCl_4_-treated cirrhotic rats have a higher expression of GC-A as compared with normal rats and suggests the close relationship between HSC activation and GC-A upregulation [36]. Our findings also revealed that ANP facilitates cGMP production and enhances PKG activity in LX-2 cells, indicating that ANP could activate GC-A/cGMP/PKG signaling in HSCs. We also elucidated that ANP suppresses proliferation while TGF-β-stimulates pro-fibrogenic activity in LX-2 cells. Recent reports have indicated that intracellular PKG negatively regulates the activation of HSCs. Franko et al. demonstrated that genomic deletion of PKG by siRNA confers the transcript profile involved in the activated state to HSCs [37]. In their study, PKG-deleted LX-2 cells promoted the upregulation of α-SMA and altered the ECM remodeling activity caused by the imbalance of MMP2/TIMP2 expression [37]. Similarly, our data show that ANP inhibited cell proliferation and TGF-β1-stimulated increases in pro-fibrotic gene expressions as well as enhanced PKG activity in LX-2 cells. Meanwhile, although treatment with SAC increased plasma CNP levels in CCl_4_-treated mice, administration of CNP did not alter the fibrotic properties in LX-2 cells due to the absence of its receptor, NPR2/GC-B. Tao et al. previously reported that human myofibroblastic HSCs expressed GC-B which is relevant to the inhibition of both the growth and contraction of HSCs as opposed to our results [21]. However, they analyzed human myofibroblastic HSCs obtained from outgrowths of normal liver explants during surgical resection for liver tumors and evaluated mRNA levels of GC-A and GC-B [21]. Therefore, further investigation is required to determine the cause of the inconsistent findings regarding the expression of NP receptors in HSCs. 

Our results show that hepatic AT1R expression and plasma AT-II levels were increased in the CCl_4_-treated mice. Thus, we assessed the direct effects of VAL on AT-II-stimulated pro-fibrogenic activity in HSCs. The AT-II/AT1R axis activates PKC, which mediates the activation of several downstream effectors, including extracellular-signal-regulated kinase 1/2, p38, and c-Jun N-terminal kinase, resulting in fibrogenic gene transcription in HSC-driven liver fibrosis [38,39]. Our previous report demonstrated that AT-II stimulates the expression of *TIMP1*, a pro-fibrogenic mediator, thereby inhibiting MMPs and protecting against activated HSC death through the AT1R/PKC signaling pathway [31]. Li et al. also reported that AT-II could directly induce the upregulation of connective tissue growth factor (CTGF), a key driver of mitogen and fibroblast chemoattractant, and ECM synthesis and secretion via AT1R/PKC activation in human HSCs [32]. Based on these findings, our results have revealed that AT-II enhances PKC activity in a dose-dependent manner but is effectively attenuated by VAL-mediated AT1R blockade. ARB-mediated inhibition of PKC activity results in the suppression of cell proliferation and the expression of *TIMP1* and *CTGF* as well as *COL1A1* in LX-2 cells. 

There are some limitations to this study. First, the results show that SAC and VAL did not modify the status of hepatocyte injury in CCl_4_-treated mice, suggesting that both agents have the potential to specifically affect HSCs. The recent literature has reported that cGMP-dependent PKG was not expressed in primary human hepatocytes [37]. Another report has also illustrated that ARBs did not modify the serum levels of liver enzymes in a murine liver fibrosis model [16]. However, to further clarify the effects of both agents on hepatocytes, additional in vitro and in vivo studies using other models would be needed. Second, a novel mechanistic insight into the NEP-dependent regulation of fibrotic properties of HSCs was recently revealed. Ortiz et al. demonstrated that NEP could degrade its substrate (neuropeptide Y [NPY]) and interfere with the NPY/NPY1 receptor axis in HSCs, resulting in the shift from an anti-fibrogenic response to a pro-fibrogenic response [40]. Thus, further studies are required to verify whether this molecular mechanism is relevant to the anti-fibrotic effects of SAC in the current experimental model. Moreover, this study has revealed the effect of SAC/VAL in preventing hepatic fibrogenesis, whereas it was not clarified whether both agents could promote hepatic fibrosis regression. Additional analyses are needed to explore the effect on hepatic fibrosis and potential liver regeneration in different liver fibrotic models.

## 5. Conclusions

In conclusion, through in vivo and in vitro experiments, a combination of NEP-i and ARB effectively prevents the progression of murine liver fibrosis and this effect is associated with the inhibition of HSC proliferation and pro-fibrogenic activity by activating ANP-dependent GC-A/cGMP/PKG signaling and inhibiting AT-II-dependent AT1R/PKC signaling. Additionally, the actions of both agents were achieved without causing hepatic and renal toxicity. Although SAC/VAL has been approved as an ANRI (Entresto) for the management of heart failure, our results suggest that this drug may eventually emerge as a viable treatment option for hepatic fibrosis.

## Figures and Tables

**Figure 1 biomedicines-11-01295-f001:**
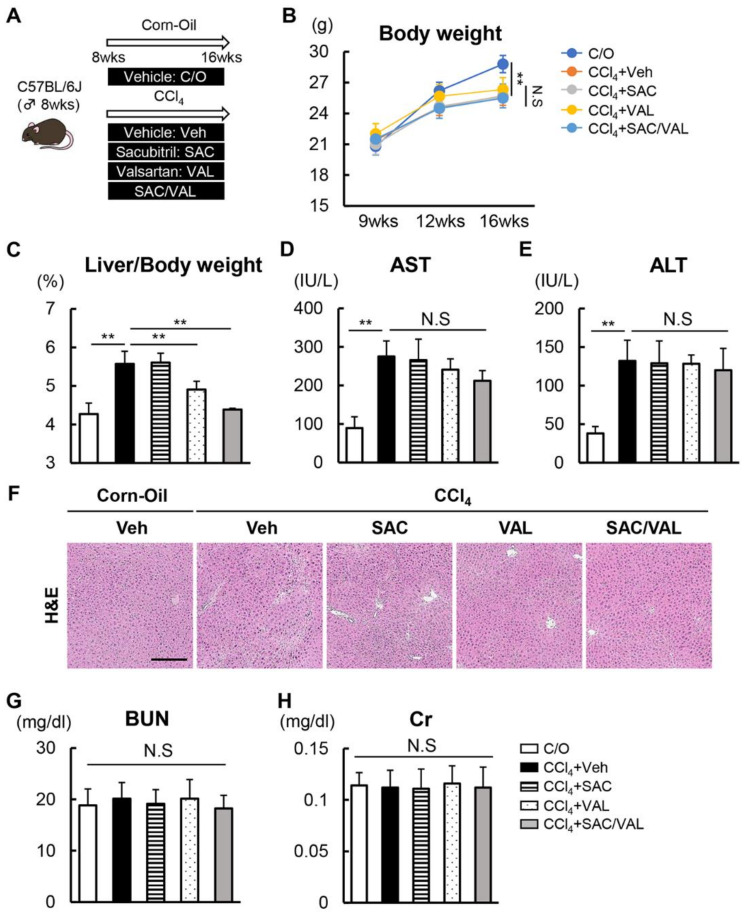
Effect of SAC/VAL on hepatic inflammation and renal function in CCl_4_-treated mice. (**A**) In vivo experimental design. (**B**) Chronological change in the body weight of the mice. (**C**) Liver/body weight. (**D**,**E**) Serum levels of (**D**) AST and (**E**) ALT at the end of the experiment. (**F**) H&E-stained liver sections. Scale bar: 50 μm. (**G**,**H**) Serum levels of (**G**) BUN and (**H**) Cr at the end of the experiment. The data are the mean ± standard deviation (*n* = 10). ** *p* < 0.01, indicating a significant difference between the groups. N.S: not significant, C/O: corn oil, CCl4: carbon tetrachloride, Veh: vehicle, SAC: sacubitril, VAL: valsartan, AST: aspartate transaminase, ALT: alanine aminotransferase, H&E: hematoxylin and eosin stain, BUN: blood urea nitrogen, and Cr: creatinine.

**Figure 2 biomedicines-11-01295-f002:**
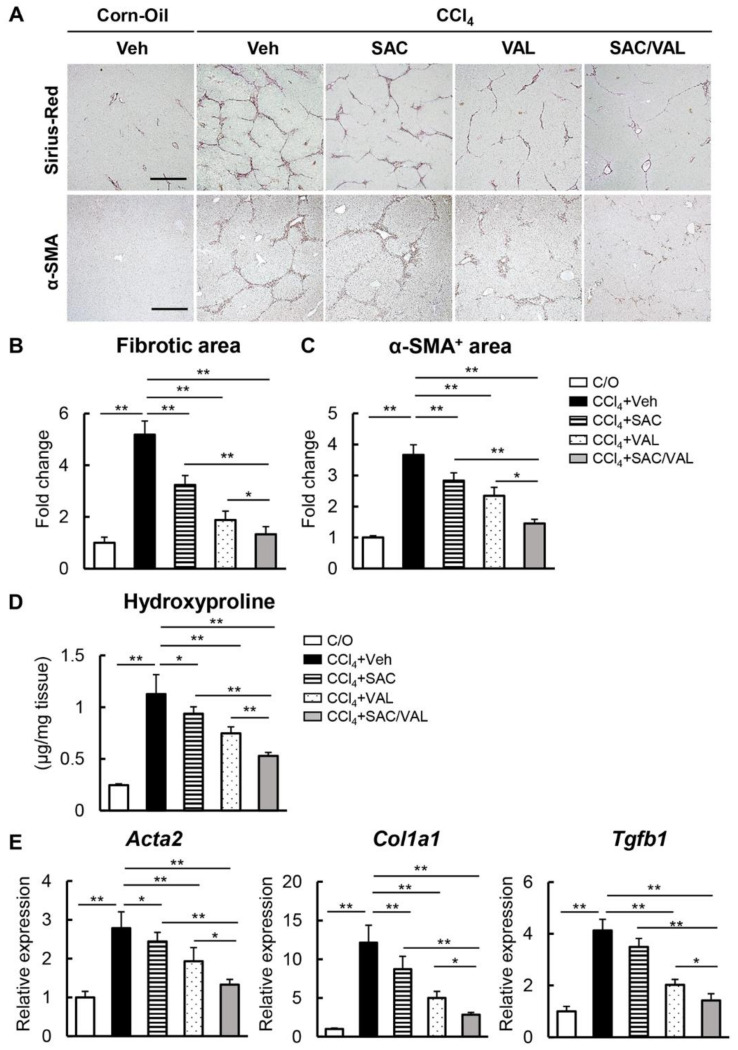
Effect of SAC/VAL on liver fibrosis development in CCl_4_-treated mice. (**A**) Sirius-Red- and α-SMA-stained liver sections. Scale bar: 50 μm. (**B**,**C**): Quantitation of (**B**) fibrotic area and (**C**) α-SMA^+^ area in the high-power field. (**D**) Hepatic concentration of hydroxyproline. (**E**) mRNA levels of *Acta2*, *Col1a1*, and *Tgfb1* in the liver tissues. Relative expression levels are indicated as fold changes to the C/O group (**B**,**C**,**E**). Data are the mean ± standard deviation (*n* = 10). * *p* < 0.05; ** *p* < 0.01, indicating a significant difference between groups. N.S: not significant, C/O: corn oil, CCl4: carbon tetrachloride, Veh: vehicle, SAC: sacubitril, VAL: valsartan, and α-SMA: α-smooth muscle actin.

**Figure 3 biomedicines-11-01295-f003:**
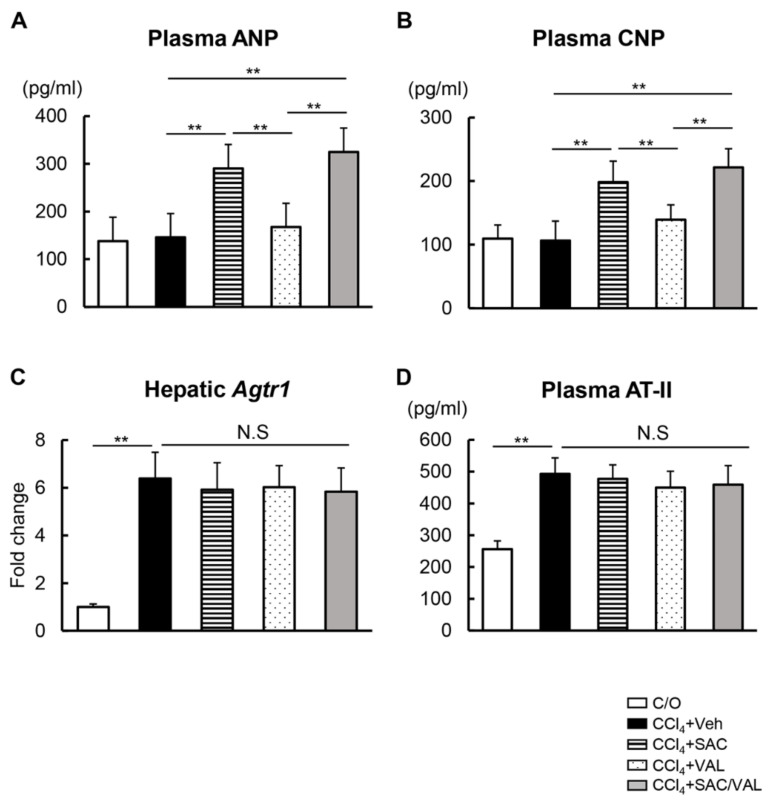
Plasma ANP, CNP, and AT-II levels and hepatic AT1R expression. (**A**) Plasma levels of ANP. (**B**) Plasma levels of CNP. (**C**) Hepatic mRNA expression levels of Agtr1. Relative expression levels are indicated as fold changes to the C/O group. (**D**) Plasma levels of AT-II. Data are the mean ± standard deviation (*n* = 10). ** *p* < 0.01, indicating a significant difference between the groups. N.S: not significant, C/O: corn-oil, Veh: vehicle, CCl4: carbon tetrachloride, SAC: sacubitril, VAL: valsartan, ANP: atrial natriuretic peptide, CNP: C-type natriuretic peptide, AT-II: angiotensin-II, and AT1R: angiotensin II receptor type 1.

**Figure 4 biomedicines-11-01295-f004:**
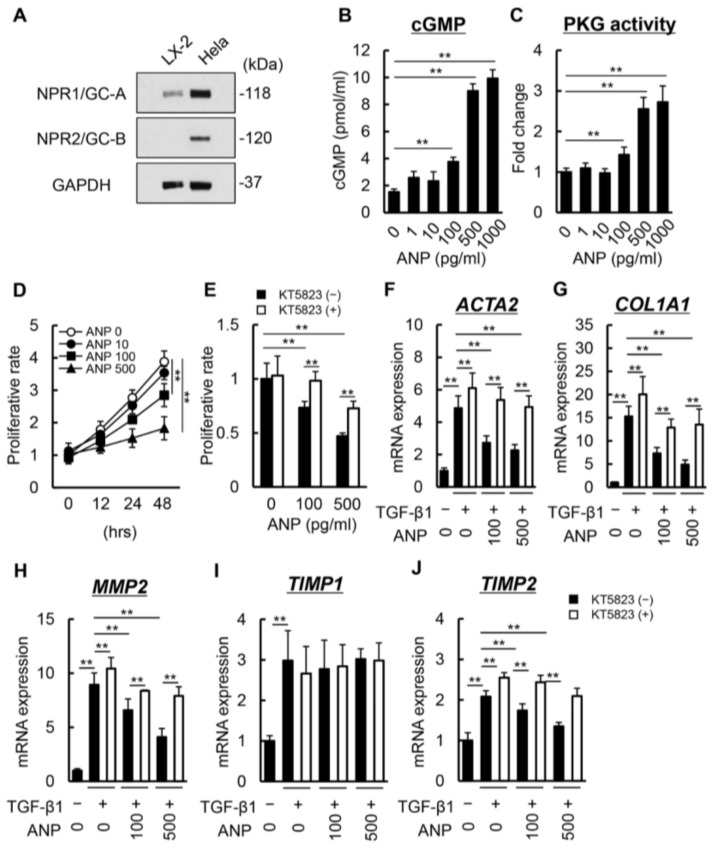
Effect of ANP on LX-2 cell proliferation and activation. (**A**) Western blots for NPR1/GC-A, NPR2/GC-B, and GAPDH protein levels in LX-2 whole-cell lysate. Hela cells were used as a positive control according to the manufacturer’s protocol. (**B**) Intracellular cGMP levels in LX-2 cells incubated with ANP (0–1000 pg/mL). (**C**) Intracellular PKG activity in LX-2 cells incubated with ANP (0–1000 pg/mL). (**D**,**E**) Cell proliferation of LX-2 cells exposed to (**D**) ANP (0–500 pg/mL) and (**E**) ANP (0, 100 and 500 pg/mL) and a PKG inhibitor, KT5823 (10 μM), for 48 h. (**F**–**J**) Relative mRNA expression of (**F**) *ACTA2*, (**G**) *COL1A1*, (**H**) *MMP2*, (**I**) *TIMP1*, and (**J**) *TIMP2* in LX-2 cells incubated with TGF-β1 (10 ng/mL), ANP (0, 100 and 500 pg/mL), and a PKG inhibitor, KT5823 (10 μM). Relative expression levels are indicated as fold changes to the values of the non-treatment groups (**C**–**J**). Data are the mean ± standard deviation (*n* = 8). ** *p* < 0.01, indicating a significant difference between the groups. ANP: atrial natriuretic peptide, NPR: natriuretic peptide receptor, GC: guanylate cyclase, GAPDH: glyceraldehyde 3-phosphate dehydrogenase, cGMP: cyclic guanosine monophosphate, PKG: protein kinase G, and TGF-β: transforming growth factor-β.

**Figure 5 biomedicines-11-01295-f005:**
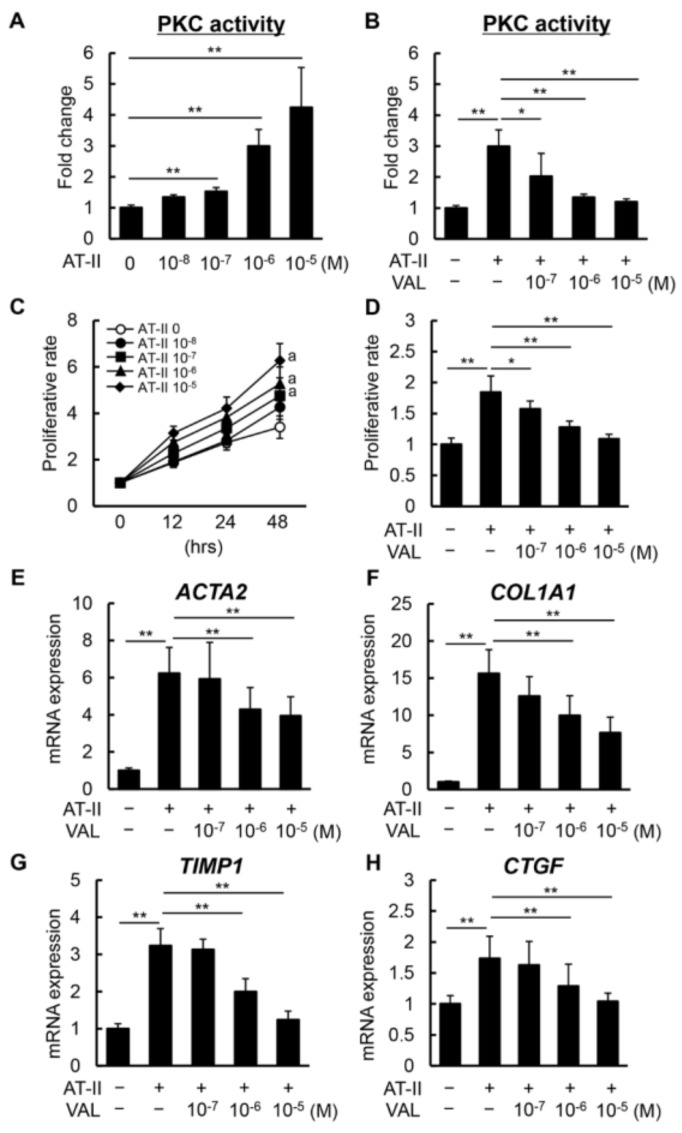
Effect of VAL on LX-2 cell proliferation and activation. (**A**) Intracellular PKC activity in LX-2 cells incubated with AT-II (0–10^−5^ M). (**B**) Intracellular PKC activity in LX-2 cells exposed to AT-II (10^−6^ M) and VAL (0–10^−5^ M). (**C**,**D**) Cell proliferation of LX-2 cells exposed to (**C**) AT-II (0–10^−5^ M) and (**D**) AT-II (10^−6^ M) and VAL (0–10^−5^ M) for 48 h. (**E**–**H**) mRNA levels of (**E**) *ACTA2*, (**F**) *COL1A1*, (**G**) *TIMP1*, and (**H**) *CTGF* in LX-2 cells exposed to AT-II (10^−6^ M) and VAL (0–10^−5^ M). Relative expression levels are indicated as fold changes to the values of the non-treatment groups (**A**–**H**). Data are the mean ± standard deviation (*n* = 8). * *p* < 0.05; ** *p* < 0.01, indicating a significant difference between the groups. ^a^ *p* < 0.01 vs. AT-II (0 M) group. AT-II: angiotensin-II, VAL: valsartan, and PKC: protein kinase C.

## Data Availability

The data that support the findings of this study are available from the corresponding author upon reasonable request.

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
