# Peer review of "A Combination of an Angiotensin II Receptor and a Neprilysin Inhibitor Attenuates Liver Fibrosis by Preventing Hepatic Stellate Cell Activation"

_biomedicines, 2023, doi:10.3390/biomedicines11051295_

Round 1

Reviewer 1 Report

Liver fibrosis is a health risk. ANP and CNP have been shown to downregulate fibrosis. In this work, the authors evaluated the effects of SAC and VAL on the ANP/CNP mediated regulation of liver fibrosis using a wide variety of methodologies that covers transcription, translation, and function. Overall this is a thorough and well-presented manuscript. I however have a few minor suggestions to improve the clarity of the work.

1. Due to the extensive use of abbreviations, it would be beneficial to add footnotes to the manuscript where all the abbreviations are in one place.

2. For section "Combination of NEP-i with ARB suppresses CCl4-induced liver fibrosis ", it would be helpful to provide a brief explanation of the rationale of choosing the physiological parameters examined, e.g. AST, ALT, etc.

3. For section "Increased ANP levels inhibit proliferative and fibrogenic activity through cGMP/PKG signaling ", it would be helpful to briefly mention why Hela cells were used here.

Author Response

We really appreciate the reviewer for positively evaluating our manuscript and offering valuable suggestions.

Reviewer 1

Liver fibrosis is a health risk. ANP and CNP have been shown to downregulate fibrosis. In this work, the authors evaluated the effects of SAC and VAL on the ANP/CNP mediated regulation of liver fibrosis using a wide variety of methodologies that covers transcription, translation, and function. Overall this is a thorough and well-presented manuscript. I however have a few minor suggestions to improve the clarity of the work.

  1. Due to the extensive use of abbreviations, it would be beneficial to add footnotes to the manuscript where all the abbreviations are in one place.

Answer

We really thank the reviewer for giving the important comment. We added the list of abbreviations used in the manuscript next page to Abstract.

  1. For section "Combination of NEP-i with ARB suppresses CCl4-induced liver fibrosis", it would be helpful to provide a brief explanation of the rationale of choosing the physiological parameters examined, e.g. AST, ALT, etc.

Answer

We appreciate the reviewer for kind comment. We showed the data of these physiological markers to elucidate that both agents did not affect hepatic inflammation in CCl4-induced liver inflammation as well as did not cause hepatic and renal toxicity. We added this explanation in the revised manuscript.

  1. For section "Increased ANP levels inhibit proliferative and fibrogenic activity through cGMP/PKG signaling ", it would be helpful to briefly mention why Hela cells were used here.

Answer

We appreciate the reviewer for kind comment. Hela cells were used as positive controls according to the manufacturer’s protocol of both antibodies for western blotting assay. We added this information in the revised manuscript.

Reviewer 2 Report

The authors have performed an excellently driven experimental study on mice which suggested a potential therapy for targeting liver fibrosis. Only a minor revision is necessary:

1. Please provide a separate conclusion chapter, which should not contain any reference to previously published work.

2. Please provide a separate ethics chapter among the material and method section.

3. Please provide more detail for each figure legend and please list the abbreviations among the figure legends.

4. Please correct English language errors

Author Response

We really appreciate the reviewer for positively evaluating our manuscript and offering valuable suggestions.

Reviewer 2

The authors have performed an excellently driven experimental study on mice which suggested a potential therapy for targeting liver fibrosis. Only a minor revision is necessary:

  1. Please provide a separate conclusion chapter, which should not contain any reference to previously published work.

Answer

According to the reviewer’s comment, we added a separate conclusion chapter in the revised manuscript.

  1. Please provide a separate ethics chapter among the material and method section.

Answer

According to the reviewer’s comment, we added a separate ethics chapter in the materials and methods part of revised manuscript.

  1. Please provide more detail for each figure legend and please list the abbreviations among the figure legends.

Answer

According to the reviewer’s comment, we rewrote figure legends in detail and added the list of abbreviations for each figure legends.

  1. Please correct English language errors

Answer

We really thank the reviewer for kind comment. According to the reviewer’s comment, we corrected English language errors.

Reviewer 3 Report

The authors present interesting results. A number of questions arose while reading it.

1. The authors need to specify what statistical criteria the authors used to find statistically significant differences.

2. The authors demonstrated reduced signs of hepatic fibrosis with sacubitril and valsartan therapy. What happens under their influence with myofibroblasts? Do they die or turn into resting stellate cells?

Author Response

We really appreciate the reviewer for positively evaluating our manuscript and offering valuable suggestions.

Reviewer 3

The authors present interesting results. A number of questions arose while reading it.

  1. The authors need to specify what statistical criteria the authors used to find statistically significant differences.

Answer

We apologize the reviewer for insufficient description. Means were compared between two groups by Student’s t-test. We added this method in the revised manuscript.

  1. The authors demonstrated reduced signs of hepatic fibrosis with sacubitril and valsartan therapy. What happens under their influence with myofibroblasts? Do they die or turn into resting stellate cells?

Answer

We really appreciate the reviewer for suggesting the important points. In our study, we aimed to examine the effect that sacubitril and valsartan could prevent liver fibrosis progression, hence experimental mice underwent the treatment with both agents from the same time point as the induction of liver fibrosis with CCl4. Sacubitril and valsartan attenuated proliferative and pro-fibrogenic potencies in activated hepatic stellate cells during fibrosis progression. However, only the result of present study is not enough to concisely answer the reviewer’s query whether both agents could induce cell apoptosis or reverse toward quiescent state in myofibroblasts which are associated with liver fibrosis regression. To accomplish this, it is necessary to further assess the effect of both agents in an established model of liver fibrosis. Since this point is very important, we discussed as a limitation.

Round 2

Reviewer 3 Report

All questions were answered satisfactorily.